# OpenReview forum: "Norm$\times$Direction: Restoring the Missing Query Norm in Vision Linear Attention"
_ICLR.cc/2026/Conference — ICLR 2026 Conference Withdrawn Submission_

### Official Review · Reviewer_bubT · 2025-11-01

**Soundness:** 3
**Presentation:** 2
**Contribution:** 3
**Rating:** 8
**Confidence:** 3

**Summary:**

This paper provides an in-depth analysis of the performance of linear transformer, which is inferior to that of vanilla transformer, and proposes that it is due to the inability to preserve the Q-norm in the process of linear attention computation, resulting in the loss of the negative correlation between the entropy of the attn matrix and the Q-norm, which leads to the excessive smoothing of the distribution of the attn matrix distribution to be excessively smooth, thus reducing the performance of the transformer.

**Strengths:**

- This paper has a unique idea, proposes a novel angle, the line is fluent and clear, and the proof process is complete.
- The paper presents its own conjecture for the problem that linear attention does not perform as well as vanilla attention, focusing on the negative correlation between the distributional spikes (entropy) of the ATTENTION MATRIX and the Query-norm, which is an interesting entry point, and there is a novelty in the motivation of the paper.
- This paper gives an ingenious, efficient injection of query-norm while preserving the computational complexity of linear attention, which takes into account the nonnegativity required by the linear attention kernel function and at the same time reconstructs the q-norm-entropy relation that is lost in linear attention, and gives a concrete reasoning process with sufficient experiments to prove its effectiveness.
- Overall, I found this article enlightening in that it gives an entry point for analyzing transformer performance and broadens the ideas for future related analyses.

**Weaknesses:**

Overall, I found this article enlightening, but I still have a few minor questions I'd like to raise:
I don't see too many problems, but I personally have a small comment, I would suggest that the q-norm/entropy correlation plots of linear vs. vanilla attention should be compared in a more forward position, so that the reader can more quickly visualize the core ideas of the paper.

**Questions:**

Also I have a couple of minor questions:
- Notice that the transformer also exhibits different performance variations when the transformer is encoded with different positions, are they also due to a change in the query-norm-entropy relationship or is it something else?
- Is the INJECTION scheme unique and are there other possible INJECTION schemes for controlling the query-norm-entropy relationship? Also, do different positional coding schemes have an impact on the INJECTION calculation process?

---

> ### Author Response · Authors · 2025-11-21
> **Response to Reviewer bubT**
>
> We sincerely appreciate the reviewer bubT’s positive assessment and encouraging scores. In the following, we present our responses and further clarifications to the points raised.
>
> > Weakness: Overall, I found this article enlightening, but I still have a few minor questions I'd like to raise: I don't see too many problems, but I personally have a small comment, I would suggest that the q-norm/entropy correlation plots of linear vs. vanilla attention should be compared in a more forward position, so that the reader can more quickly visualize the core ideas of the paper.
>
> Thank you for your helpful suggestions. We agree that presenting this comparison earlier could help readers grasp the core idea of our method more quickly. In the revised manuscript, we will move these plots to a more prominent and forward position and adjust the narrative flow accordingly to improve readability and highlight the key intuition.
>
> >Q1. Notice that the transformer also exhibits different performance variations when the transformer is encoded with different positions, are they also due to a change in the query-norm-entropy relationship or is it something else. Q2. Also, do different positional coding schemes have an impact on the INJECTION calculation process?
>
> Thank you for your question. When a Transformer uses different positional encoding schemes, what changes is how the attention mechanism (whether standard or linear) computes the **similarity** between queries and keys. Positional encodings introduce a **position-dependent bias** so that even if two tokens are identical (i.e., the same q–k pair), their similarity scores differ when they appear at different positions.
> Therefore, we consider that the observed performance differences are not caused by changes in the query-norm–entropy relationship, but rather by how effectively the positional encoding captures absolute or relative position information.
> Furthermore, common positional encoding methods, such as absolute encodings (sinusoidal or learnable) and relative encodings (RoPE, cosformer), can all be integrated into our method. We will further explore this aspect in future work.
>
>
> >Q2. Is the INJECTION scheme unique and are there other possible INJECTION schemes for controlling the query-norm-entropy relationship?
>
> Thank you for your question. The Injection scheme is not unique. Our proposed design directly follows from the theoretical analysis in the paper, and therefore we adopt a power function for simplicity and efficiency, which explicitly makes the entropy become query-norm aware. Also, other alternatives are also possible. For instance, one could replace the power function with other spiky functions such as exp(), where the query norm can be explicitly incorporated into the exponential term to achieve a similar effect.
> In this work, we characterize the fundamental relationship between query norm and attention entropy, a property that has been overlooked in prior linear-attention formulations, and we propose an approach to recover this behavior. Building on this theoretical insight, we will further investigate a broader family of Injection schemes in future work.
>
> We hope the above clarifications address your concerns. Please feel free to let us know if you have any further questions or suggestions.

---

> > ### Comment · Reviewer_bubT · 2025-11-28
> >
> > That's good, i'll keep my rating, This article is still very inspiring to me.

---

> > > ### Author Response · Authors · 2025-11-28
> > >
> > > Thank you for your thoughtful feedback and encouraging words. We sincerely appreciate your time and are glad that the revisions and clarification have addressed your concerns. We will further add these updates to the final version of our paper.
> > >
> > > Authors

---

### Official Review · Reviewer_dBiZ · 2025-11-01

**Soundness:** 3
**Presentation:** 3
**Contribution:** 3
**Rating:** 4
**Confidence:** 3

**Summary:**

This manuscript introduces NaLaFormer, a novel linear attention mechanism addressing the expressiveness gap between linear and softmax attention by leveraging norm×direction (ND) decomposition of query and key vectors. It restores the query norm’s role in regulating attention distribution spikiness through a norm-aware feature map and preserves non-negativity without information loss via a cosine-based direction similarity metric. Validated across multi-modal tasks, NaLaFormer achieves state-of-the-art performance—including up to 7.5% accuracy gain on ImageNet-1K, 4.7% mIoU improvement on ADE20K, and 92.3% peak memory reduction in super-resolution—while outperforming baselines like Mamba on language tasks and the LRA benchmark.

**Strengths:**

1. This paper resolves two core limitations of linear attention (query norm cancellation and destructive non-negativity enforcement) through theoretically grounded ND decomposition, which is novel to me.
2. The experiment are conducted on competitive benchmarks, e.g., ImageNet, COCO, ADE20K and DIV2K.
3. The formula and figure are clear and well-illustrated.

**Weaknesses:**

1. How the proposed linear attention work in diffusion is not clear. Since existing works show that linear attention can also perform well in diffusion transformers[1], it is encouraged to add some DiT experiments.
2. The RoPE is used in the proposed method, but not discussed.
3. The ablation study in λ and τ is missing.

[1] Efficient Diffusion Transformer with Step-wise Dynamic Attention Mediators, in ECCV 2024.

**Questions:**

1. Does NaLaFormer need infra optimization when serving?

---

> ### Author Response · Authors · 2025-11-21
> **Response to Reviewer dBiZ**
>
> We thank the reviewer dBiZ for the detailed review and insightful comments, which help us improve the clarity and quality of the manuscript. In the following, we provide our responses and clarifications regarding the weaknesses and questions raised.
> >W1. How the proposed linear attention work in diffusion is not clear. Since existing works show that linear attention can also perform well in diffusion transformers, it is encouraged to add some DiT experiments.
>
> Thank you very much for the helpful suggestion. We agree that Diffusion Transformers provide a suitable setting for evaluating the effectiveness of linear attention. Following the work you recommended, we conducted experiments on diffusion transformer S/2 to further validate our approach, and the results are shown below.
>
> |Model|FID $\downarrow$|sFID $\downarrow$|IS $\uparrow$|Precision $\uparrow$|Recall $\uparrow$|
> |----|----|----|----|----|----|
> |DiT|68.40|-|-|-|-|
> |DiG [2]|62.06|11.77|22.81|0.39|0.56|
> |NaLaDiT (ours)|61.64|12.40|23.24|0.40|0.58|
>
> |Model|FID $\downarrow$|sFID $\downarrow$|IS $\uparrow$|Precision $\uparrow$|Recall $\uparrow$|
> |----|----|----|----|----|----|
> |SiT|58.61|9.25|24.31|0.41|0.59|
> |Efficient SiT [1]|53.57|9.01|27.26|0.43|0.61|
> |NaLaSiT (ours)|53.08|8.94|27.63|0.43|0.62|
>
> [1] Pu, Yifan, et al. "Efficient diffusion transformer with step-wise dynamic attention mediators." ECCV, 2024.
>
> [2] Zhu, Lianghui, et al. "Dig: Scalable and efficient diffusion models with gated linear attention." CVPR, 2025.
>
> The experimental results show that our method can be effectively adapted to diffusion transformer models. Compared with other linear-attention approaches applied in diffusion transformers, our algorithm achieves substantial performance improvements.
> We will include these results in the revised manuscript and appropriately cite the work you kindly pointed us to.
>
> >W2. The RoPE is used in the proposed method, but not discussed.
>
> Thank you for highlighting this import aspect.
> Existing works, such as Cosformer [1] and RoPE [2], keep part of the information with trigonometric functions by using cosine-based functions. Cosformer [1] replaces the softmax in attention with a cosine-based distance metric, using cosine similarity to directly measure query-key alignment, while RoPE [2] encodes absolute positions via rotation matrices in complex space to represent the relative position.
> However, both kinds of cosine similarity are employed for **positional decay**, which differs from our cosine-based method targeting the **similarity and dimensions with opposite signals**, shown as follows:
>
> \begin{equation}
>     \operatorname{SM}_{\operatorname{cosformer}}(\mathbf{q}_n,\mathbf{k}_m)=\phi(\mathbf{q_n})\phi(\mathbf{k_m})^\top \underbrace{\cos(\frac{(m-n)\pi}{2M})}\_{\text{relative position}},
> \end{equation}
>
> \begin{equation}
> \operatorname{SM}_{\operatorname{rope}}(\mathbf{q}_n,\mathbf{k}_m)=\phi(\mathbf{q_n})\underbrace{R\_{\Theta,n-m}^d}\_{\text{relative position}}\varphi(\mathbf{k_m})^\top
> \end{equation}
>
> \begin{equation}
> \operatorname{SM}_{\operatorname{ours}}(\mathbf{q},\mathbf{k})=\sum\_{i=1}^{d} \underbrace{cos(\phi(\mathbf{q})_i-\phi(\mathbf{k})_i)}\_{\text{dimensional cosine similarity}}
> \end{equation}
>
> The discussion will be updated correspondingly in the revised manuscript.
>
> [1] Qin, Zhen, et al. "cosFormer: Rethinking Softmax In Attention." ICLR, 2022.
>
> [2] Su, Jianlin, et al. "RoFormer: Enhanced transformer with Rotary Position Embedding." Neurocomputing. 2024.
>
> >W3. The ablation study in $\lambda$ and $\tau$ is missing.
>
> Thank you for your question. We conduct the **ablation study** on both image classification (CV) and document retrieval (NLP) from LRA benchmark. The results are as follows,
>
> $\lambda$|$\tau$|Retrieval|Image
> ----|----|----|----
> 3|0.5|80.42|44.54
> 5|0.5|80.17|41.91
> 7|0.5|80.13|40.73
> 3|1|80.13|41.80
> 3|2|80.35|42.12
>
> The experimental results demonstrate the impact of $\tau$, which controls the fluctuation range of the query norm, and $\lambda$, which regulates the base-level spikiness, on the model performance. We will further explore norm-aware feature map designs in future work and  include these results in the revised manuscript.
>
> >Q1. Does NaLaFormer need infra optimization when serving?
>
> Thank you for your question regarding the infrastructure optimization of NaLaFormer. Due to the nature of linear attention, NaLaFormer is already friendly for inference, which do not rely on a KV cache and instead, compresses contextual information into a fixed-size hidden state. Therefore, no additional system-level or serving-time optimizations were required in our implementation. For on-device or resource-constrained deployment, techniques such as quantization may be necessary to further improve efficiency. We will consider these directions highly relevant and explore them in future work.
>
> We hope the above clarifications address your concerns. Please feel free to let us know if you have any further questions or suggestions.

---

> ### Author Response · Authors · 2025-11-23
> **Kindly Reminder: Please let us know whether we have addressed all the issues.**
>
> Dear Reviewer dBiZ,
>
> Thank you very much for taking the time to review our paper, and your insights are greatly appreciated. We have carefully addressed the questions and comments raised in your review, and we hope our responses adequately address your concerns. Please let us know if there are any remaining issues that need further clarification, or if you require additional details. We are happy to provide further explanations as needed.
>
> Thank you,
>
> Authors

---

> ### Author Response · Authors · 2025-11-26
> **Follow-up on Response to Reviewer Comments**
>
> Dear Reviewer dBiZ,
>
> I hope this message finds you well. I wanted to follow up on our previous correspondence regarding the revisions to our manuscript. We truly appreciate your valuable feedback, and we have made every effort to address all the points raised in your review.
>
> If you’ve had the chance to review our revised manuscript and responses, we would be grateful for any further comments or confirmation that the issues have been fully resolved. Should you need any additional information or clarification, please do not hesitate to let us know.
>
> Thank you once again for your time and effort in reviewing our work. We look forward to your response.
>
> Best regards,
>
> Authors of paper 3255

---

> ### Comment · Reviewer_dBiZ · 2025-11-28
>
> I appreciate the effective rebuttal regarding the weaknesses. However, regarding serving, I still have a little bit concern. Since most popular linear attention method still have its accelerated version (see www.github.com/fla-org/flash-linear-attention), could you discuss if your method supports hardware-aware acceleration strategies to further facilitate practical deployment?

---

> > ### Author Response · Authors · 2025-11-28
> > **Response to Reviewer dBiZ**
> >
> > Thank you for taking the time to read our response and for raising this new question. We appreciate our response have addressed most of your concerns. Regarding your new question on hardware-aware acceleration strategies, we provide the following clarifications:
> >
> > Our method primarily operates on the feature map **before the linear attention computation**. Following the norm-aware spiky mapping and the cosine-based transformation of the model inputs, attention is computed using **the standard linear attention formulation**, which naturally enables integration with Triton operator optimizations developed for FLA. However, since FLA is primarily designed for causal attention, for vision tasks we instead use [fbi_la](https://github.com/fla-org/flash-bidirectional-linear-attention), a library also maintained by the FLA team, which provides optimized implementations for bidirectional linear attention through triton. Given that our algorithm consists purely of matrix multiplications, it can directly leverage the Triton implementation of simple linear attention. Furthermore, the pure matrix-multiplication structure is highly compatible with quantization techniques (e.g., the INT8 quantization used in SANA [1]), leaving substantial room for further optimization, thus our method is more friendly with quantization method. Building on our current approach, we plan to explore hardware-level optimizations in future work.
> >
> > Based on the fbi_la implementation, we measured the memory usage and runtime of our NaLaFormer-T model on 8 NVIDIA RTX 3090 GPUs, under a batch size of 32 and logging every 10 steps. The results are summarized below.
> >
> > ||Time|max memory|
> > |---|---|---|
> > |PyTorch|0.2617|8110|
> > |fla|0.2357 (save 10\%)|7753 (save 5\%)|
> >
> > We hope the above clarifications address your concerns. Please feel free to let us know if you have any further questions or suggestions.
> >
> > [1] Xie, Enze, et al. "Sana: Efficient high-resolution image synthesis with linear diffusion transformers." arXiv preprint arXiv:2410.10629 (2024).

---

> > ### Author Response · Authors · 2025-11-30
> > **Response to Reviewer dBiZ**
> >
> > Thank you for taking the time to read our response and for raising this new question. We appreciate our response have addressed most of your concerns. Regarding your new question on hardware-aware acceleration strategies, we provide the following clarifications:
> >
> > Our method primarily operates on the feature map **before the linear attention computation**. Following the norm-aware spiky mapping and the cosine-based transformation of the model inputs, attention is computed using **the standard linear attention formulation**, which naturally enables integration with Triton operator optimizations developed for FLA. However, since FLA is primarily designed for causal attention, for vision tasks we instead use [fbi_la](https://github.com/fla-org/flash-bidirectional-linear-attention), a library also maintained by the FLA team, which provides optimized implementations for bidirectional linear attention through triton. Given that our algorithm consists purely of matrix multiplications, it can directly leverage the Triton implementation of simple linear attention. Furthermore, the pure matrix-multiplication structure is highly compatible with quantization techniques (e.g., the INT8 quantization used in SANA [1]), leaving substantial room for further optimization, thus our method is more friendly with quantization method. Building on our current approach, we plan to explore hardware-level optimizations in future work.
> >
> > Based on the fbi_la implementation, we measured the memory usage and runtime of our NaLaFormer-T model on 8 NVIDIA RTX 3090 GPUs, under a batch size of 32 and logging every 10 steps. The results are summarized below.
> >
> > ||Time|max memory|
> > |---|---|---|
> > |PyTorch|0.2617|8110|
> > |fla|0.2357 (save 10\%)|7753 (save 5\%)|
> >
> > We hope the above clarifications address your concerns. Please feel free to let us know if you have any further questions or suggestions.
> >
> > [1] Xie, Enze, et al. "Sana: Efficient high-resolution image synthesis with linear diffusion transformers." arXiv preprint arXiv:2410.10629 (2024).

---

### Official Review · Reviewer_2qdC · 2025-11-02

**Soundness:** 2
**Presentation:** 3
**Contribution:** 2
**Rating:** 2
**Confidence:** 4

**Summary:**

This paper introduces NaLaFormer, a Norm-Aware Linear Attention mechanism for Transformer models. It targets two major weaknesses in conventional linear attention: the disregard for query norm information and the loss of expressive negative inner-product interactions due to enforced non-negativity. The authors propose a decomposition of query and key vectors into norm and direction components, allowing dynamic entropy reduction modulated by query norms and a norm-preserving cosine mapping that enforces non-negativity while retaining directional richness. Substantial theoretical analysis is provided regarding the role of query norms in attention entropy, and empirical evaluations show performance gains on vision and language modeling benchmarks.

**Strengths:**

1. To avoid negative values, the re-mapped cosine direction is a somewhat clever technical contribution.

2. The experiment results show strong improvements over many baselines. On ImageNet-1K, NaLaFormer variants outperform recent linear attention and some softmax-based ViT models across all model sizes, often by substantial margins.

3. Both vision (understanding and super-resolution) and language tasks are included, which extends the breadth of the applications of this paper.

**Weaknesses:**

1. The paper does not cite or compare to MetaLA [1] and InLine [2], which both focus on matching softmax’s spikiness or optimizing the linear approximation. Empirical comparisons and deeper methodological discussion are crucial, especially in language tasks, to establish both novelty and superiority.

2. The exact model definitions and training details are not fully disclosed in the appendix, such as Swish activation before the classifier, Layerscales, 1024-dim classifier, convolution patch embedding, RoPE, and CPE, which is a convolution bypass in the attention module. Some or many of the baselines lack these enhanced designs; therefore, a clean version of the proposed method should be carefully ablated.

3. The cosine direction works similarly to rotary position embeddings (RoPE) [3] in large language and vision models; this paper does not discuss the technical insights between the proposed method and RoPE.

[1] Chou, Yuhong, Man Yao, Kexin Wang, Yuqi Pan, Rui-Jie Zhu, Jibin Wu, Yiran Zhong, Yu Qiao, Bo Xu, and Guoqi Li. "MetaLA: Unified optimal linear approximation to softmax attention map." Advances in Neural Information Processing Systems 37 (2024): 71034-71067.

[2] Han, Dongchen, Yifan Pu, Zhuofan Xia, Yizeng Han, Xuran Pan, Xiu Li, Jiwen Lu, Shiji Song, and Gao Huang. "Bridging the divide: Reconsidering softmax and linear attention." Advances in Neural Information Processing Systems 37 (2024): 79221-79245.

[3] Su, Jianlin, Murtadha Ahmed, Yu Lu, Shengfeng Pan, Wen Bo, and Yunfeng Liu. "Roformer: Enhanced transformer with rotary position embedding." Neurocomputing 568 (2024): 127063.

**Questions:**

I have no other questions about this submission.

---

> ### Author Response · Authors · 2025-11-21
> **Response to Reviewer 2qdC (1)**
>
> We thank the reviewer 2qdC for the detailed review and insightful comments, which help us improve the clarity and quality of the manuscript. In the following, we provide our responses and clarifications regarding reviews.
>
> >W1. The paper does not cite or compare to MetaLA and InLine, which both focus on matching softmax’s spikiness or optimizing the linear approximation. Empirical comparisons and deeper methodological discussion are crucial, especially in language tasks, to establish both novelty and superiority.
>
> Thank you for your suggestion. MetaLA constructs a lightweight recurrent-form linear attention by defining the optimal linear approximation conditions of the softmax attention map. However, this method is primarily designed for **language models with the causal strategy**. When applied to encoder architectures, such as ViT models or bidirectional attention, the model performance becomes sensitive to the **scanning order**, making it less suitable for vision tasks.
> Our method is mainly tailored for Vision Transformers. To evaluate the performance and generalization ability of our proposed feature map, we only conducted decoder-only LLM experiments by replacing the feature map module. We will include a discussion of this work and explore further optimization specifically for decoder-only computations in future research.
>
> Inline provides an important insight by proving that the softmax function is injective in most cases, whereas linear attention is not. By modifying the normalization scheme, it restores the injectivity of linear attention. In addition, Inline introduces a local-attention residual (a convolution module) to enhance local bias, thereby compensating for softmax’s strong capability in modeling local patterns. This work highlights the importance of injectivity in linear attention and uses vectors with identical norms but different directions as counterexamples to address this limitation. However, Inline overlooks the relationship between attention distribution uncertainty and the query/key norms—an essential property of the softmax function.
>
> **We will incorporate the above discussion and cite both works in the Related Work section of the revised manuscript. Meanwhile, we will add a clear comparison between our method and InLine (for Vision Transformers) on Swin-T, and we will include MetaLA’s ImageNet-1K results in the ImageNet-1K evaluation table. The results are shown in W2 section.**
>
> >W2. The exact model definitions and training details are not fully disclosed in the appendix, such as Swish activation before the classifier, Layerscales, 1024-dim classifier, convolution patch embedding, RoPE, and CPE, which is a convolution bypass in the attention module. Some or many of the baselines lack these enhanced designs; therefore, a clean version of the proposed method should be carefully ablated.
>
> Thank you for your suggestion. The ImageNet classification experiments are conducted on top of the current sota method, RALA [1]. To ensure a fair comparison with the RALA baseline, we follow its model design. In order to verify that the performance gains of our method indeed stem from the advanced linear-attention mechanism, we further include the following ablation studies under the XT-size settings: blocks [2, 2, 4, 2], channels [32, 64, 192, 384], and heads [1, 2, 6, 12]. The results indicate that the influence of these components on model performance is limited, thereby further demonstrating the superiority of our norm-aware linear attention.
>
> Comparison with the sota efficient models.
>
> |Model|Agent-Deit-T$_{ECCV24}$|Local-Vim-T$_{ECCV24}$|MetaLA$_{NIPS24}$|Mambaout-F$_{CVPR25}$|EfficientVMamba-S$_{AAAI25}$|Ours-XT|
> |----|----|----|----|----|----|----|
> |Params|6M|8M|6M|7M|11M|8M|
> |FLOPs|1.2G|1.5G|-|1.2G|1.3G|1.0G|
> |Acc|74.9%|76.2%|75.3%|78.9%|78.7%|**79.1%**|
>
> Ablation studies.
>
> |w.o.|Rope|CPE|Layerscales|Swish|Ours-XT|
> |----|----|----|----|----|----|
> |Acc| 79.1%|78.9%|79.1%|78.7%|79.1%|

---

> > ### Author Response · Authors · 2025-11-21
> > **Response to Reviewer 2qdC (2)**
> >
> > Furthermore, in table 9 of our work, we conducted the experiments by **only integrating our attention mechanism into the Swin Transformer** as the **clean and fair comparison**, which is widely used as a standard backbone for evaluating linear-attention methods [2][3][4][5], and we additionally included InLine-swin-t as a baseline in Table 9 (revised in Table 10).
> >
> > |Linear Attn|Swin|Efficientvit$_{ICCV23}$[6]|FLatten$_{ICCV23}$[2]|Agent$_{ECCV24}$[3]|Inline$_{NIPS24}$[5]|Pola$_{ICLR25}$[4]|ours|
> > |----|----|----|----|----|----|----|----|
> > |Accuracy|81.3%|81.8%|82.1%|82.6%|82.4%|82.6%|82.9%|
> >
> > We will add all these results into our main table, and supply the ablation study in the revised manuscript.
> >
> > [1] Fan, Qihang, et al. "Breaking the low-rank dilemma of linear attention." CVPR, 2025.
> >
> > [2] Han, Dongchen, et al. "Flatten transformer: Vision transformer using focused linear attention." ICCV, 2023.
> >
> > [3] Han, Dongchen, et al. "Agent attention: On the integration of softmax and linear attention." ECCV, 2024.
> >
> > [4] Meng, Weikang, et al. "PolaFormer: Polarity-aware Linear Attention for Vision Transformers." ICLR, 2025.
> >
> > [5] Han, Dongchen, et al. "Bridging the divide: Reconsidering softmax and linear attention." NeurIPS, 2024.
> >
> > [6] Cai, Han, et al. "Efficientvit: Lightweight multi-scale attention for high-resolution dense prediction." ICCV, 2023.
> >
> > >W3. The cosine direction works similarly to rotary position embeddings (RoPE) in large language and vision models; this paper does not discuss the technical insights between the proposed method and RoPE.
> >
> > Thank you for highlighting this import aspect.
> > Existing works, such as Cosformer [1] and RoPE [2], keep part of the information with trigonometric functions by using cosine-based functions. Cosformer [1] replaces the softmax in attention with a cosine-based distance metric, using cosine similarity to directly measure query-key alignment, while RoPE [2] encodes absolute positions via rotation matrices in complex space to represent the relative position.
> > However, both kinds of cosine similarity are employed for **positional decay**, which differs from our cosine inhibition method targeting the **similarity and dimensions with opposite signals**, shown as follows:
> >
> > \begin{equation}
> >    \operatorname{SM}_{\operatorname{cosformer}}(\mathbf{q}_n,\mathbf{k}_m)=\phi(\mathbf{q_n})\phi(\mathbf{k_m})^\top \underbrace{\cos(\frac{(m-n)\pi}{2M})}\_{\text{relative position}},
> > \end{equation}
> >
> > \begin{equation}
> >   \operatorname{SM}_{\operatorname{rope}}(\mathbf{q}_n,\mathbf{k}_m)=\phi(\mathbf{q_n})\underbrace{R\_{\Theta,n-m}^d}\_{\text{relative position}}\varphi(\mathbf{k_m})^\top,
> > \end{equation}
> >
> > \begin{equation}
> >   \operatorname{SM}_{\operatorname{ours}}(\mathbf{q},\mathbf{k})=\sum\_{i=1}^{d} \underbrace{cos(\phi(\mathbf{q})_i-\phi(\mathbf{k})_i)}\_{\text{dimensional cosine similarity}}.
> > \end{equation}
> >
> > [1] Qin, Zhen, et al. "cosFormer: Rethinking Softmax In Attention." International Conference on Learning Representations. 2022.
> > [2] Su, Jianlin, et al. "RoFormer: Enhanced transformer with Rotary Position Embedding." Neurocomputing. 2024.
> >
> > We hope the above clarifications address your concerns. Please feel free to let us know if you have any further questions or suggestions!

---

> ### Author Response · Authors · 2025-11-23
> **Kindly Reminder: Please let us know whether we have addressed all the issues.**
>
> Dear Reviewer 2qdC,
>
> Thank you very much for taking the time to review our paper, and your insights are greatly appreciated. We have carefully addressed the questions and comments raised in your review, and we hope our responses adequately address your concerns. Please let us know if there are any remaining issues that need further clarification, or if you require additional details. We are happy to provide further explanations as needed.
>
> Thank you,
>
> Authors

---

> ### Author Response · Authors · 2025-11-26
> **Follow-up on Response to Reviewer Comments**
>
> Dear Reviewer 2qdC,
>
> I hope this message finds you well. I wanted to follow up on our previous correspondence regarding the revisions to our manuscript. We truly appreciate your valuable feedback, and we have made every effort to address all the points raised in your review.
>
> If you’ve had the chance to review our revised manuscript and responses, we would be grateful for any further comments or confirmation that the issues have been fully resolved. Should you need any additional information or clarification, please do not hesitate to let us know.
>
> Thank you once again for your time and effort in reviewing our work. We look forward to your response.
>
> Best regards,
>
> Authors of paper 3255

---

### Author Response · Authors · 2025-11-21
**A Summary of Reviews**

Thank you to all reviewers for taking the time to review our paper and providing valuable and constructive comments! We are very grateful to all reviewers for giving our paper recognition and in the following aspects:

1.  **Clever technical contribution and novel insight:** The re-mapped cosine direction is a somewhat clever technical contribution; theoretically grounded ND decomposition, which is novel to me; This paper has a unique idea, proposes a novel angle (all reviewers).

2. **Complete and novel theoretical support** (reviewer dBiZ and bubT).

3. **Clarity and presentation quality:** The formula and figure are clear and well-illustrated; the line is fluent and clear, and the proof process is complete. (reviewer dBiZ and bubT).

4. **Generality and Extensive  experiments with impressive results:**  Both vision (understanding and super-resolution) and language tasks are included, which extends the breadth of the applications of this paper; gives a concrete reasoning process with sufficient experiments to prove its effectiveness (all reviewers).

5. This paper gives an entry point for analyzing transformer performance and broadens the ideas for future related analyses. (reviewer bubT).

Once again, I extend my heartfelt thanks to all the reviewers for your invaluable feedback on our paper!
We supplemented this paper with a few additional experiments and works, including：

1. Add the image classification tasks with another XT size.

2. Discuss about the difference between RoPE and the proposed method; two exemplary related works worth referencing.

3. Ablation studies about the hyper-parameters and components.

4. Add some DiT experiments and baseline models.

You can find our individual responses below your review comments. **If you have any more concerns or questions, we are entirely open to continuing the discussion with you!**

---

### Author Response · Authors · 2025-11-30
**[For New AC] Summary of Reviewer Comments and Our Rebuttal**

Dear ICLR 2026 AC,

Thank you for your effort and time. For your convenience, we have summarized the reviewers’ concerns along with the main points of our rebuttal as follows.

- **1. Difference between RoPE and the proposed cosine-based direction similarity (Reviewer 2qdC, dBiZ).**

    - We provide a detailed discussion and the final derivation of the similarity formulation.
    - We clarify that although both methods use cosine similarity, RoPE focuses on encoding ***relative positional information***, whereas our approach is designed to compute ***dimensional cosine similarities***.
    - **Reviewer dBiZ has already confirmed that our rebuttal has resolved the concerns.**

- **2. Add DiT experiments (Reviewer dBiZ).**

    - Following the work as the reviewer recommended, we conducted experiments on ***diffusion transformer, size S/2*** on both SiT and DiT to further validate our approach.
    - The results show that our method can be ***successfully adapted to diffusion transformer models***. Compared with other linear-attention approaches applied in diffusion transformers, our algorithm achieves ***consistent performance improvements in DiT***.
    - **Reviewer dBiZ has already stated that our rebuttal has resolved the concerns.**

- **3. Ablation studies about the hyperparameter and the impact of each component on image classification tasks (Reviewer 2qdC, dBiZ).**

    - We clarify that our implementation strictly follows the current SOTA baselines RALA; therefore, all of these settings are ***kept consistent with the sota baseline***.
    - Moreover, none of these components are used in our SR, language modeling, or DiT experiments.
    - To address the reviewers’ concerns, we additionally ***conducted additional ablation studies to examine the contribution of each component.***
    - The results show that the influence of these components on model performance is **limited**, thereby further demonstrating the superiority of our norm-aware linear attention.
    - We also conduted the ***ablations about the hyperparameter*** $\lambda$ and $\tau$.
    - **Reviewer dBiZ has already confirmed that our rebuttal has resolved the concerns.**

- **4. Discussion about two related works, MetaLA and InLine. (Reviewer 2qdC)**

    - We ***cite and discussed*** the properties of both works, and incorporate the discussion into the Related Work section of the revised manuscript.
    - We ***add a clear comparison*** between our method and InLine (for Vision Transformers) on Swin-T, and we include MetaLA’s ImageNet-1K results in the ImageNet-1K evaluation table in the revised manuscript to show our method achieves consistent improvement.

- **5. Infra optimization when serving, and whether supports hardware-aware acceleration strategies, fla (Reviewer dBiZ).**

    - We present an analysis demonstrating that ***our method naturally supports integration with the Triton acceleration*** as developed in FLA.
    - Since FLA is mainly designed for causal attention, for vision tasks we instead use [fbi_la](https://github.com/fla-org/flash-bidirectional-linear-attention), a library also maintained by the FLA team, which provides optimized implementations for bidirectional linear attention through triton.
    - We also ***made the comparison the speed and memory between pytorch and triton implemetation.***

- **6.  Is the INJECTION scheme unique and are there other possible INJECTION schemes for controlling the query-norm-entropy relationship? (Reviewer bubT)**

    - We detailed clarify that we adopt a power function for simplicity and efficiency, which ***explicitly makes the entropy become query-norm aware***.
    - We provide some more practical INJECTION schemes.
    - Highlight our insight that  ***the fundamental relationship between query norm and attention entropy***, a property that has been overlooked in prior linear-attention formulations.
    - **The reviewer bubT indicated that our rebuttal satisfactorily addressed their concerns and mentioned that this article is very inspiring.**

- **7. Supplementary Experiments in the Revised Manuscript.**

    - DiT-S/2 and SiT-S/2 integrated with the proposed NaLaFormer.
    - A new configuration of NaLaFormer for image classification task, to enable fair comparison with the baseline models MetaLA and InLine, as suggested by reviewer 2qdC.
    - Ablation studies on the hyperparameter $\lambda$ and $\tau$.
    - Ablation studies on components in the image classification task.
    - Speed and memory comparisons between the PyTorch implementation and the Triton implementation.

We hope this consolidated summary facilitates a more efficient and convenient review of our paper.

Best regards,

Authors of paper 3255

---

### Author Response · Authors · 2025-11-30
**[For New AC] Overview of Reviewer Engagement During the Rebuttal Period**

Dear ICLR 2026 Area Chair,

We appreciate your time handling our submission. We would like to bring a few facts to your attention so you can make a fair assessment of the discussion that followed our rebuttal.

By the rebuttal revert time, two of the three reviewers (dBiZ and bubT) responded to our rebuttal and explicitly stated that ***our responses addressed most of their concerns***. In particular, reviewer bubT wrote ***highly supportive comments***, for example:

>“This paper has a ***unique idea, proposes a novel angle***; the line is fluent and clear, and the proof process is complete; Overall, I found this article ***enlightening in that it gives an entry point*** for analyzing transformer performance and broadens the ideas for future related analyses... This article is still ***very inspiring to me***.”

Additionally, We have answered additional questions raised by these two reviewers in our follow-up comments.

***However, reviewer 2qdC has not replied to our rebuttal and has provided no further engagement.*** We note that reviewer 2qdC assigned a low score (2'),  but several of the concerns appear to arise from misunderstandings of our method (the differences between our approach and RoPE) rather than from substantive high-level issues (e.g., novelty, motivation, insight, or generalizability) and the points raised are largely similar to the minor clarifications requested by reviewer dBiZ, ***who has since indicated that our rebuttal adequately addressed these concerns***. The remaining comments from reviewer 2qdC mainly relate to detail-level issues (citations, discussions, and additional ablations), ***which does not seem fully aligned with the low score (2') assigned***, and there has been ***no further interaction despite our follow-up responses.***

Based on this fact, ***we would greatly appreciate it if you would take a moment to review the discussion thread with reviewer 2qdC*** when forming the final judgement. In particular, we hope that the positive follow-up from the other two reviewers, together with the fact that reviewer 2qdC did not raise substantive high-level issues, may help provide a more complete picture. **If reviewer 2qdC provides any further response at a later stage, please feel free to share it with us and we would be more than happy to follow up and clarify any remaining questions.**

Thank you very much for your attention and for handling the review process.

Best regards,

Authors of paper 3255

---

### Note · Authors · 2026-01-29

I have read and agree with the venue's withdrawal policy on behalf of myself and my co-authors.

---

### Meta-Review · Area_Chair_gr2W · 2026-01-09

**Summary:**

It received ratings of 2,4,8. Authors provided rebuttal.

The authors wrote that 2 of the reviewers are convinced by the response. One of them (rating of 4) wrote "I appreciate the effective rebuttal regarding the weaknesses. However, ..." and the other one (rating of 8) which was already positive wrote "That's good, i'll keep my rating, This article is still very inspiring to me."

**Reviewer Concerns:**

The main concerns are: insufficient related work comparison, incomplete model and training details, unclear relationship to RoPE, missing ablation studies, the diffusion setting is unclear, comparisons may be unfair because some or many of the baselines lack these enhanced designs.

The rebuttal addresses some of these concerns, but I do not think it addresses them all to be enough to push the paper to the accept bucket. For instance the rebuttal discusses the relationship to RoPE, but given their similarity, the fact that the original submission does not even cite RoPE is still a valid concern.

**Reviewer Scores:**

2,4,8.

2: did not reply to rebuttal and I do not think they would have been fully convinced by it. For instance the fact that RoPE is similar and not even discussed or cited is still a valid concern.
4: replied by writing "I appreciate the effective rebuttal regarding the weaknesses. However, ...". It is not clear if they wanted to raise their score.
8: replied positively and said will keep the score.

---

### Decision · Program_Chairs · 2026-01-26

Reject